# Close Associations of Gum Bleeding with Systemic Diseases in Late Adolescence

**DOI:** 10.3390/ijerph17124290

**Published:** 2020-06-16

**Authors:** Masanobu Abe, Akihisa Mitani, Atsushi Yao, Hideyuki Takeshima, Liang Zong, Kazuto Hoshi, Shintaro Yanagimoto

**Affiliations:** 1Division for Health Service Promotion, The University of Tokyo, Tokyo 113-0033, Japan; mitania-int@h.u-tokyo.ac.jp (A.M.); yaoa-int@h.u-tokyo.ac.jp (A.Y.); yanagimoto@hc.u-tokyo.ac.jp (S.Y.); 2Department of Oral & Maxillofacial Surgery, The University of Tokyo Hospital, Tokyo 113-8655, Japan; hoshi-ora@h.u-tokyo.ac.jp; 3Division of Epigenomics, National Cancer Center Research Institute, Tokyo 104-0045, Japan; hitakesh@ncc.go.jp (H.T.); zl20014111@163.com (L.Z.)

**Keywords:** gum bleeding, periodontal disease, gingivitis, periodontitis, late adolescence, adolescent, systemic disease

## Abstract

Background: Though it is well known that periodontal diseases are associated with various systemic diseases in adults, the associations in late adolescents have not been adequately elucidated. We investigated the association between gum bleeding (a major symptom of periodontal diseases) and common systemic diseases in late adolescents: allergic, respiratory, and otorhinolaryngologic diseases. Methods: We conducted a retrospective review of the mandatory medical questionnaires administered as a part of legally required freshman medical checkup between April 2017 and April 2019 at the University of Tokyo. Among the total of 9376 sets of responses, 9098 sets from students aged less than 20 were analyzed. An χ2 test and univariate and multivariate binomial logistic regression analyses were performed using SAS ver. 9.4. A value of *p* < 0.05 was accepted as significant. Results: According to the questionnaire data, 3321 students (36.5%; 2780 males and 541 females) responded that they experienced gum bleeding whenever they brushed their teeth. These students had significantly higher incidence rates of otitis media/externa and asthma/cough-variant asthma (*p* = 0.001 and *p* = 0.006, respectively). The results of the multivariate analysis showed significant rates of the following complications among these students: (1) otitis media/externa (odds ratio (OR) 1.691; 95% confidence interval (CI): 1.193–2.396; *p* = 0.003), (2) asthma/cough-variant asthma (OR 1.303; 95% CI: 1.091–1.556; *p* = 0.003), and (3) male gender (OR 1.536; 95% CI: 1.337–1.765; *p* < 0.001). Conclusions: Gum bleeding was closely associated with otitis media/externa and asthma in late adolescents. Our study reinforces new evidence about the association between periodontal diseases and asthma, and it reveals a novel and close association between gum bleeding and otitis media/externa.

## 1. Background

Periodontal disease, which consists of gingivitis and periodontitis, comprises a wide range of inflammatory conditions in the oral cavity. It is highly prevalent in adults, and its severity increases with age. Global data have demonstrated that (1) mild-to-moderate periodontitis affects most adults and (2) severe periodontitis is the sixth-most prevalent health condition, affecting 10.8% of the population (743 million) worldwide [1,2,3]. Importantly, periodontal diseases are not just the major cause of tooth loss [4,5,6], they are also associated with various systemic diseases, including cardiovascular disorders [7,8], rheumatism [9,10], diabetes mellitus [11,12], and respiratory disorders [13]. Therefore, there has been an increasing research attention worldwide on the relationship between oral health and overall health [14,15,16].

The majority of studies investigating the relationship between periodontal diseases and systemic diseases have been focused on adults—primarily middle-aged and elderly people. In contrast, the same has not been well investigated in younger people, and, therefore, the relationship has yet not been clarified in late adolescence. Though severe periodontitis accompanied by tooth mobility is quite rare in late adolescence, studies have reported that gingivitis (which can cause gum bleeding) is occasionally observed at that age [17,18]. Considering this situation, we conducted the present study to investigate the relationship between gum bleeding and the following common systemic diseases, which are often observed in late adolescents: allergic diseases/disorders (pollinosis, food/drug allergy, inhaled antigen allergy, and atopic dermatitis), otorhinolaryngologic diseases (allergic rhinitis, otitis media/externa, and sinusitis), and respiratory disorders (infectious diseases, pneumothorax/mediastinal emphysema, and asthma/cough-variant asthma). We believe that if critical associations are identified in late adolescents, this knowledge would not only contribute to the treatment of late adolescence-specific diseases but also to the prevention of diseases emerging later in adulthood.

## 2. Methods

### 2.1. Study Design and Population

We performed a retrospective review of the mandatory medical questionnaires administered as a part of legally required freshman medical checkup between April 2017 and April 2019 at the University of Tokyo. The questionnaire was self-administered, closed-, and open-ended. Among the total of 9376 responses (7563 males and 1813 females aged 17–59 years; mean age of 18.4 years), a total of 9098 sets of responses from students aged less than 20 years (7316 males and 1782 females aged 17–19 years; mean age of 18.3 years) were subjected to analyses.

### 2.2. Questionnaire

A self-administered, closed-, and open-ended questionnaire was distributed to the university students. The gum health status was evaluated by using the yes/no question “Are you aware of gum bleeding when you brush your teeth?” (symptom of periodontal disease). When a student reported observing gum bleeding during toothbrushing, he/she was considered to have periodontal disease (gingivitis or periodontitis). The medical history of the student was assessed using the option “Please choose the presence or absence of the history of diseases/disorders.” When a student chose the presence of a history for diseases/disorders, he/she was asked to write the specific name(s) of the disease(s)/disorder(s).

After tallying each student’s responses on the questionnaire, the associations between gum bleeding and allergic diseases/disorders, otorhinolaryngologic diseases, and respiratory diseases in late adolescence was investigated. Diseases/disorders identified less than 30 cases were excluded from the analysis. Disease-related medication was not investigated in the questionnaire.

### 2.3. Statistical Analyses

Statistical analysis was performed using the χ2 test. A multivariate analysis was conducted using a binomial logistic regression model. A value of *p* < 0.05 (two-sided) was accepted as statistically significant. The statistical software programs IBM SPSS Statistics ver. 21.0 (IBM Corp., Armonk, NY, USA) and SAS ver. 9.4 (SAS Institute Inc., Cary, NC, USA) were used in this study.

### 2.4. Ethical Approval

This study was approved by the research ethics committee of the University of Tokyo in 2018; approval no. 18–197 (currently revised as no. 19-324).

## 3. Results

### 3.1. Proportion of Students with Awareness of Gum Bleeding

The questionnaire was distributed to 9376 university students (7563 males and 1813 females aged 17–59 years; mean age of 18.4 years) who entered the University of Tokyo in April 2017, 2018, and 2019. Then, the data obtained from 9098 students aged <20 years (7316 males and 1782 females aged 17–19 years; mean age of 18.3 years) were analyzed in this study. Of these 9098 students, 3321 (36.5%) responded that they were aware of bleeding from their gums when they brushed their teeth. Because gum bleeding is one of the symptoms of periodontal disease, we categorized those respondents with this awareness as suffering from periodontal disease. As shown in Table 1, a significantly higher percentage of male students (38.0%; *n* = 2780) reported experiencing gum bleeding than female students (30.4%; *n* = 541) (*p* < 0.001).

### 3.2. Prevalence of Common Systemic Diseases in Late Adolescence

As presented in Table 2, the prevalence rates of each of the queried common systemic diseases among the complete series of respondents were as follows: allergic diseases/disorders: pollinosis (15.57%; *n* = 1417), food/drug allergy (3.03%; *n* = 276), inhaled antigen allergy (2.02%; *n* = 184), and atopic dermatitis (7.03%; *n* = 640); otorhinolaryngologic diseases: allergic rhinitis (15.08%; *n* = 1372), otitis media/externa (1.98%; *n* = 180), and sinusitis (1.59%; *n* = 145); and respiratory diseases: infectious diseases (0.68%; *n* = 62), pneumothorax/mediastinal emphysema (1.22%; *n* = 111), and asthma/cough-variant asthma (9.60%; *n* = 873).

### 3.3. Associations between Gum Bleeding and Common Systemic Diseases in Late Adolescence

As summarized in Table 2, the students who reported experiencing gum bleeding had significantly higher incidence rates of a history of otitis media/externa (*p* = 0.001) and asthma/cough-variant asthma (*p* = 0.006). There were no significant incidences of the other diseases/disorders, as follows: pollinosis (*p* = 0.181), food/drug allergy (*p* = 0.325), inhaled antigen allergy (*p* = 0.197), atopic dermatitis (*p* = 0.449), allergic rhinitis (*p* = 0.071), sinusitis (*p* = 0.785), respiratory infectious diseases (*p* = 0.069), and pneumothorax/mediastinal emphysema (*p* = 0.689).

In the present study, the analysis of only the male respondents showed that those who reported experiencing gum bleeding also had higher incidence rates of the history of otitis media/externa and asthma/cough-variant asthma (*p* = 0.028 and *p* = 0.006, respectively). The female respondents who reported experiencing gum bleeding had a significant incidence of otitis media/externa (*p* = 0.005) but not of asthma/cough-variant asthma (*p* = 0.928). In particular, female respondents with gum bleeding had a significant incidence of respiratory infectious diseases.

A multivariate analysis was conducted for the complete series of respondents using a binomial logistic regression model; gum bleeding was the objective variable (gum bleeding as an event), and the above-mentioned 10 diseases/disorders (pollinosis, food/drug allergy, inhaled antigen allergy, atopic dermatitis, allergic rhinitis, otitis media/externa, sinusitis, respiratory infectious diseases, pneumothorax/mediastinal emphysema, and asthma/cough-variant asthma) plus gender were the explanatory variables (Table 3). The results demonstrated that gum bleeding was closely associated with otitis media/externa (odds ratio (OR) 1.691; 95% confidence interval (CI): 1.193–2.396; *p* = 0.003), asthma/cough-variant asthma (OR 1.303; 95% CI: 1.091–1.556; *p* = 0.003), and male gender (OR 1.536; 95% CI: 1.337–1.765; *p* < 0.001).

## 4. Discussion

Oral diseases constitute a major global public health problem, affecting more than 3.5 billion people [1,2]. Periodontal disease, a chronic and progressive inflammatory disorder, is one of the most common oral diseases, like dental caries [19]. Periodontal disease affects the supporting structure of the teeth (i.e., the gingiva, periodontal ligaments, and bones), and severe periodontitis causes tooth loss [4,5,6]. Most importantly, periodontal diseases are known to be associated with various systemic diseases, including respiratory disorders [13], diabetes mellitus [11,12], cardiovascular disorders [7,8], rheumatism [9,10], systemic infection [20,21,22,23,24], metabolic syndrome [25], and preterm birth [26] in adults, primarily middle-aged and elderly people.

In late adolescents, although severe periodontitis is quite rare, studies have reported that gingival inflammation (gingivitis) is often observed [17,18]. However, as the association between periodontal diseases and systemic diseases has not been adequately elucidated in late adolescents, we focused on the relationship between gingival health status and common systemic diseases in late adolescents. Our study results demonstrated that otitis media/externa was closely associated with gum bleeding in late adolescents (*p* = 0.001). The multivariate analysis showed that otitis media/externa was an independent risk factor for gum bleeding, with a high OR of 1.691 (95% CI: 1.193–2.396; *p* = 0.003). To our knowledge, this is the first study to demonstrate a relationship between periodontal disease and otitis media/externa.

Though there is less knowledge regarding the association between periodontal diseases and otitis media/externa, bacterial translocation might be involved as a possible mechanism of the association [14]. In fact, oral bacteria that cause otitis media have been reported. Kakuta et al. described a case of severe acute otitis media that was caused by *Campylobacter rectus* [27]. Concerning other otorhinolaryngologic diseases, an association between allergic rhinitis and periodontal diseases has been suggested [28], and in the present study, although this association was not statistically significant, the results demonstrated a similar tendency.

In the present study, we also found that gum bleeding was closely associated with the history of asthma in late adolescence among the university students (*p* = 0.006). The multivariate analysis identified asthma as an independent risk factor for gum bleeding (OR 1.303; 95% CI: 1.091–1.556; *p* = 0.003). Earlier investigations have indicated that people with asthma are more likely to suffer from gingivitis and periodontitis than healthy controls [29,30]. A recent meta-analysis also provided additional evidence about the close association between periodontal diseases and asthma, wherein patients with asthma demonstrated higher levels of gingival inflammation than healthy individuals [31]. Our study adds new insights into these findings, that is the association can be observed even when the population is limited to late adolescents. Several mechanisms that may contribute to the relationship between periodontal diseases and respiratory diseases, such as the aspiration of biofilm and the hematogenous dissemination of oral pathogens or inflammatory chemical mediators from the periodontal pockets, have been proposed [32]. Once this relationship is established, periodontal diseases may indirectly contribute to the recurrence and worsening of respiratory attacks. Studies have reported that antiasthmatic medication might also be associated with the interaction of diseases [33,34]. In this context, inhaled corticosteroids (which have been recommended as first-line treatment) and open-mouth breathing have been suggested to decrease saliva production, cause changes in pH, and increase biofilm accumulation [35,36]. This could be due to the immunosuppressive effect of corticosteroids that may have some influence on the response of periodontal tissues by inhibiting the host response [37].

Our multivariate analysis showed that male gender was an independent risk factor for gum bleeding, with a high OR value of 1.536 (95% CI: 1.337–1.765, *p* < 0.001). A systematic review of the literature and meta-analyses have provided broad-based evidence indicating that males are at a greater risk of developing destructive periodontal disease than females [38,39]. Though the age at which this difference would start manifesting has not been clearly elucidated, our study suggests that it would be the early stage of life. The gender difference regarding oral health consciousness/behaviors has been considered to be deeply involved in the gingival health status in late adolescence [17].

Studies have proposed that pulmonary function can be improved by the treatment and maintenance of gingival health [40,41]. Based on a report of oral bacteria that cause otitis media/externa, it has been reported that otitis media/externa is preventable by maintaining a good oral health status [27]. Though oral diseases have been separated from the mainstream healthcare system in Japan, it is essential to raise awareness among the dental and medical communities to counteract and improve the incidence of systemic diseases from the early stage of life.

In summary, our study reinforces new evidence about the association between periodontal diseases and asthma, and it reveals a novel and close association between gingivitis and otitis media/externa. However, additional studies are needed to verify these findings and to clarify the mechanism underlying the associations.

## 5. Conclusions

Gum bleeding, a major symptom of gingivitis, was found to be closely associated with the history of otitis media/externa and asthma in late adolescence. Our study reinforces new evidence about an association between periodontal diseases and asthma, and it provides the first demonstration of a close association between gum bleeding and otitis media/externa. However, further research is needed to verify these findings and to clarify the mechanism underlying the associations.

## 6. Ethics Approval and Consent to Participate

This study was approved by the Research Ethics Committee of the University of Tokyo in 2018, approval no. 18-197 (currently revised as no. 19-324), “Retrospective analyses of medical and health record information retained by the division for health service promotion, the University of Tokyo.” We abide by all relevant laws, regulations, and the University rules for privacy. The announcement on our privacy policy is shown on the website (http://www.hc.u-tokyo.ac.jp/). In accordance with the condition stated in the ethics approval, we have posted a notice on the website of the department to announce the privacy policy and opt-out (https://www.lifescience.mext.go.jp/files/pdf/n2181_01.pdf).

## Figures and Tables

**Table 1 ijerph-17-04290-t001:** Proportion of students with gum bleeding among the 9098 late adolescents.

Status of Gum Bleeding		All (*n* = 9098)		Male (*n* = 7316)		Female (*n* = 1782)	
		*n*	%	*n*	%	*n*	%
Gum bleeding	Presence	3321	36.5	2780	38.0	541	30.4
	Absence	5777	63.5	4536	62.0	1241	69.6

**Table 2 ijerph-17-04290-t002:** Association between gum bleeding and medical history.

Medical History		All			Male			Female		
		Gum Bleeding		*p*	Gum Bleeding		*p*	Gum Bleeding		*p*
		*n* (%)			n (%)			n (%)		
		Presence	Absence		Presence	Absence		Presence	Absence	
All	*n*	3321 (100.0)	5777 (100.0)		2780 (100.0)	4536 (100.0)	-	541 (100.0)	1241 (100.0)	
Allergic diseases/disorders										
Pollinosis	1417	540 (16.26)	877 (15.18)	0.181	448 (16.12)	678 (14.95)	0.19	92 (17.01)	199 (16.04)	0.66
Food/drug allergy	276	109 (3.28)	167 (2.89)	0.325	96 (3.45)	130 (2.87)	0.18	13 (2.40)	37 (2.98)	0.6
Inhaled antigen allergy (except pollinosis)	184	76 (2.29)	108 (1.87)	0.197	63 (2.27)	83 (1.83)	0.227	13 (2.40)	25 (2.01)	0.731
Atopic dermatitis	640	243 (7.32)	397 (6.87)	0.449	209 (7.52)	320 (7.05)	0.486	34 (6.28)	77 (6.20)	1
Otorhinolaryngologic diseases										
Allergic rhinitis	1372	531 (15.99)	841 (14.56)	0.071	466 (16.76)	682 (15.04)	0.053	65 (12.01)	159 (12.81)	0.697
Otitis media/ externa	180	87 (2.62)	93 (1.61)	0.001 *	70 (2.52)	79 (1.74)	0.028 *	17 (3.14)	14 (1.13)	0.005 *
Sinusitis	145	55 (1.66)	90 (1.56)	0.785	46 (1.65)	76 (1.68)	1	9 (1.66)	14 (1.13)	0.489
Respiratory diseases										
Respiratory infectious diseases	62	30 (0.90)	32 (0.55)	0.069	21 (0.76)	26 (0.57)	0.426	9 (1.66)	6 (0.48)	0.026 *
Pneumothorax/mediastinal emphysema	111	38 (1.14)	73 (1.26)	0.689	38 (1.37)	69 (1.52)	0.665	0 (0.00)	4 (0.32)	0.437
Asthma/cough variant asthma	873	356 (10.72)	517 (8.95)	0.006 *	319 (11.47)	429 (9.46)	0.006 *	37 (6.84)	88 (7.09)	0.928

*: <0.05.

**Table 3 ijerph-17-04290-t003:** Multivariate analysis for the association between gum bleeding and medical history.

Medical History	Odds Ratio (95% Confidence Interval)	*p*
Allergic diseases/disorders		
Pollinosis	0.994 (0.859–1.151)	0.939
Food/drug allergy	1.065 (0.789–1.437)	0.68
Inhaled antigen allergy (except pollinosis)	1.253 (0.866–1.814)	0.231
Atopic dermatitis	1.030 (0.834–1.271)	0.785
Otorhinolaryngologic diseases		
Allergic rhinitis	1.084 (0.935–1.256)	0.285
Otitis media/externa	1.691 (1.193–2.396)	0.003 *
Sinusitis	0.971 (0.651–1.449)	0.885
Respiratory diseases		
Respiratory infectious diseases	1.850 (0.952–3.593)	0.07
Pneumothorax/mediastinal emphysema	1.000 (0.595–1.681)	1
Asthma/cough-variant asthma	1.303 (1.091–1.556)	0.003 *
Gender		
Male	1.536 (1.337–1.765)	<0.001*

*: *p* < 0.05.

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
