# Peer review of "Close Associations of Gum Bleeding with Systemic Diseases in Late Adolescence"

_ijerph, 2020, doi:10.3390/ijerph17124290_

Round 1
Reviewer 1 Report
The manuscript submitted to IJERPH entitled “Close Associations of Gum Bleeding with Systemic Diseases/Disorders in Adolescence” is an original article which aim to investigate possible associations between periodontal diseases and systemic diseases in adolescents.
On my opinion the article is interesting, well written, with good English. The content of the manuscript is very interesting. The authors showed possible association of gum bleeding with otitis media/externa and asthma/cough-variant asthma in a student population of Japan.
However, I highlighted some critical issues.
Title. The presence of the slash (/) in the title is not suitable.
Abstract. Not adequate in presenting a summary of the study.
Introduction. Are there studies in the literature concerning diseases in the student population? Better specify the objectives and methods of the study.
Discussion. Are there other similar studies that have shown similar results in the adult population?
Conclusion. I would be more cautious in defining the associations between the conditions analyzed in this study. Further studies are needed to confirm any association.
References. Move the square brackets before punctuation.
I don't feel qualified to judge about the English language and style.
There is no conflict of interest between me and any of the authors.
Author Response
Response to Reviewer #1
Comments:
- The manuscript submitted to IJERPH entitled “Close Associations of Gum Bleeding with Systemic Diseases/Disorders in Adolescence” is an original article which aim to investigate possible associations between periodontal diseases and systemic diseases in adolescents. On my opinion the article is interesting, well written, with good English. The content of the manuscript is very interesting. The authors showed possible association of gum bleeding with otitis media/externa and asthma/cough-variant asthma in a student population of Japan.
We appreciate this reviewer's precise and constructive comments.
- The presence of the slash (/) in the title is not suitable.
Reflecting this comment, we removed (/) and modified the title.
- Not adequate in presenting a summary of the study.
Owing to the comment, we checked and modified expression of conclusion.
- Are there studies in the literature concerning diseases in the student population? Better specify the objectives and methods of the study.
We thank the reviewer for this comment. The objectives and methods were specified owing to the comment.
- Is it possible to insert a copy of the questionnaire as supplementary material?
We prepared an additional file for the questionnaire, although it was originally written in Japanese.
- Specify the year of approval of the ethics committee. Is it a university or hospital ethics committee?
This study was approved by the research ethics committee of the University of Tokyo in 2018, approval no. 18-197. This description was added to Method section.
- Have students who claimed to have gingival bleeding been diagnosed with periodontal disease?
As the reviewer pointed out, the students are not diagnosed as periodontal diseases. Therefore, we carefully corrected inadequate words through the manuscript.
- Are there other similar studies that have shown similar results in the adult population?
Earlier investigations have indicated that people (no limitation for age) with asthma are more likely to suffer from periodontal diseases than healthy controls [31, 32]. However, no studies could be found which have shown similar results for the association between periodontal diseases and otitis in adult population.We described it
- I would be more cautious in defining the associations between the conditions analyzed in this study. Further studies are needed to confirm any association.
As the reviewer pointed out, further studies are needed to confirm the association which was found in this study. This description was added into Conclusion section.
- Move the square brackets before punctuation.
We thank the reviewer for this comment. We corrected the positions of the brackets.
Reviewer 2 Report
Age range is not correct; authors investigate people aging 17-19, but adolescents, as defined by WHO, are people between ages 10 and 19. Therefore, the paper, that aims to investigate correlation between systemic disorders and gengivitis in adolescents, has got a lack of ages (10-16). In addition, having a wide sample, authors could have include "young people" which refers to individuals between ages 10 and 24. So, the title is not correct.
Questionnaire should provided. It is not clear if the questionnaire also investigates taken disease-related drugs.
References are not exaustive. Discussion should better deep the correlation between and systemic disease/medications. It is not clear if questionnaire investigates also related disease- drugs.
Author Response
Response to Reviewer #2
Comments:
- Age range is not correct; authors investigate people aging 17-19, but adolescents, as defined by WHO, are people between ages 10 and 19. Therefore, the paper, that aims to investigate correlation between systemic disorders and gengivitis in adolescents, has got a lack of ages (10-16). In addition, having a wide sample, authors could have include "young people" which refers to individuals between ages 10 and 24. So, the title is not correct.
We appreciate this precise comment. In the revised manuscript, "late adolescents" is used instead of "adolescents" through the manuscript.
- Questionnaire should provided.
We prepared an additional file for the questionnaire, although it was originally written in Japanese.
- It is not clear if the questionnaire also investigates taken disease-related drugs.
Disease-related medication was not investigated in the questionnaire. This information was added into Method section in the revised manuscript.
- References are not exaustive.
We thank the reviewer for this comment. An important article [39] was added in to the manuscript.
- Discussion should better deep the correlation between and systemic disease/medications. It is not clear if questionnaire investigates also related disease- drugs.
As the reviewer pointed out, the effect of medicine is considered to be important for the relation of the diseases. Especially, inhaled corticosteroids might be deeply involved in periodontal health status. This was discussed in Discussion section. We would like to address this issue in our future studies.
Reviewer 3 Report
It's article is a good papers, but must be improved the part of the conclusions.
Author Response
Response to Reviewer #3
Comment:
- It's article is a good papers, but must be improved the part of the conclusions.
Thank you very much. Owing to the comment, we checked and modified expression of conclusion.
Reviewer 4 Report
Dear Authors,
the manuscript has points of innovation in the field of oral health and scientific merit.
I recommend including a "study limitations" paragraph in the discussion.
Regarding the revised manuscript, I asked the authors to include in the "discussion" section a paragraph informing the "limitations of the study".
I believe that in future studies the authors may include contextual and individual variables related to socioeconomic status, access to oral health services and oral hygiene in the model to expand the hypotheses for the results found.
Regards
Reviewer
Author Response
Response to Reviewer #4
Comments:
- the manuscript has points of innovation in the field of oral health and scientific merit.
We appreciate this reviewer's constructive comments.
- I recommend including a "study limitations" paragraph in the discussion.
Regarding the revised manuscript, I asked the authors to include in the "discussion" section a paragraph informing the "limitations of the study".
A paragraph of study limitations was included in the Discussion section.
- I believe that in future studies the authors may include contextual and individual variables related to socioeconomic status, access to oral health services and oral hygiene in the model to expand the hypotheses for the results found.
Thank you so much for the constructive comment. We will include contextual and individual variables related to socioeconomic status, access to oral health services and oral hygiene in our future studies.
Round 2
Reviewer 1 Report
The authors edited the manuscript following the reviewers' suggestions. In my opinion it could now be suitable for publication.Author Response
Thank you so much. Our manuscript was improved by the reviewer's constructive comments.
Reviewer 2 Report
Although the revision, modifying topic on LATE adolescents, discussion is still focused on adolescents in general. Is there in literature difference among age range? (i.e: early adolescents, late adolescent?). Can author explain and discuss differences?
Authors modified the age (now is late adolescents as suggested) but discussion and concepts remain the same.. Please, focus on selected age range.
Author Response
We appreciate this precise comment. Reflecting this comment, several references and confusing words were removed from the manuscript to make our concept and discussion consistent. We confirmed that the manuscript focused on late adolescent throughout, although some references dealt with wide age range.